# The Circumstances of Children with and without Disabilities or Significant Cognitive Delay Living in Ordinary Households in 30 Middle- and Low-Income Countries

Eric Emerson [1,2,3,*] and Gwynnyth Llewellyn [2]

1   Centre for Disability Research, Faculty of Health and Medicine, Lancaster University, Lancaster LA1 4YW, UK
2   Centre for Disability Research and Policy, Faculty of Medicine and Health, University of Sydney, Sydney 2006, Australia; gwynnyth.llewellyn@sydney.edu.au
3   College of Nursing and Health Sciences, Flinders University, Adelaide 5042, Australia
*   Correspondence: eric.emerson@lancaster.ac.uk

**Abstract:** Home-based early intervention is a key component of strategic approaches to preventing the loss of developmental potential among young children in middle- and low-income countries. We undertook secondary analysis of nationally representative data collected in 30 middle- and low-income countries during Round 6 (2017-) of UNICEF's Multiple Indicator Cluster Surveys. Our analyses, involving over 100,000 children with/without disabilities or significant cognitive delay, indicated that children with disabilities and/or significant cognitive delay were more likely than their peers to: be living in relative household poverty and in rural areas; have a mother with only primary level education; live in households where intimate partner violence was considered acceptable by the child's mother; be less likely to be receiving any pre-school education; have a mother with disabilities; receive low levels of parental stimulation; be exposed to violent parental discipline. For many of these indicators of poorer home circumstances, the level of risk was notably higher for children with significant cognitive delay than for children with disabilities. Our study findings suggest the need to consider tailored, evidence-based approaches to reduce and (potentially) overcome the increased risks that young children with disabilities and young children with significant cognitive delay experience in middle- and low-income countries.

**Keywords:** disability; children; families; middle-income countries; low-income countries





## 1. Introduction

Home-based early intervention programs have been identified as a key component of strategic approaches to preventing the loss of developmental potential among young children in middle- and low-income countries [1–4]. They have also been specifically identified as a key component of approaches to promoting the rights and well-being of young children with disabilities or developmental delay [5–11].

It is important that family-based interventions take account of family circumstances. For example, family circumstances such as levels of poverty and literacy may have an impact on the effectiveness of home-based early intervention programs unless they are specifically tailored to the level of resources available. As recommended by the World Health Organization and UNICEF, 'service providers must work closely with families to design and implement interventions that are culturally appropriate and meet their needs' ([5], p. 28).

The existing literature suggests that the home circumstances of children with disabilities or developmental delay may differ from those of their non-disabled/delayed peers in some important ways. For example, evidence suggests that children with disabilities and/or developmental delays in middle- and low-income countries are more likely than their peers to live in poverty, experience lower levels of parental stimulation,

suffer from undernutrition and be more likely to be exposed to violence [12–18]. However, much of this information is dated and little is known about the extent to which these risks may vary between country economic classification groups (upper-middle income, lower-middle-income, low-income).

The aim of the present paper is to provide contemporary information on the home circumstances of children under five years of age with/without disabilities/significant cognitive delay growing up in a range of upper-middle-income, lower-middle-income and low-income countries.

## 2. Experimental Section

We undertook secondary analysis of nationally representative data collected in Round 6 (2017-) of UNICEF's Multiple Indicator Cluster Surveys (MICS6) [19]. Following approval by UNICEF, MICS6 data were downloaded from http://mics.unicef.org/ (accessed on 16 July 2021). MICS6 contains several questionnaire modules. Data used in the present paper were extracted from two household modules, the module applied to all children under five living in the household and the module applied to all women aged 16–49 living in the household [20]. In order to address statistical problems associated with the clustering of children within households and facilitate the extraction of data from different modules, our target population consisted of the first enumerated child under 5 years of age in each household.

All participating countries used cluster sampling methods to derive samples representative of the national population of mothers and young children. Specific details of the sampling procedure used in each country are available in Country Reports available at http://mics.unicef.org/ (accessed on 16 July 2021). At the end of the download period (31 March 2021) nationally representative data containing disability information for children aged 2–4 were available from 30 countries (11 upper-middle-income, 11 lower-middle-income, 8 low-income). Information on these surveys (region, year of survey, response rates, sample sizes) is presented in Table 1.

**Table 1.** Surveys included in the analyses.

| Country | Region | Year of Survey | Under 5 Response Rate | Sample Size |
|---|---|---|---|---|
| Upper-Middle-Income | | | | |
| Costa Rica | LAC | 2018 | 86.9% | 2137 |
| Montenegro | ECA | 2018/19 | 60.7% | 634 |
| Cuba | LAC | 2019 | 98.5% | 3141 |
| Turkmenistan | ECA | 2019 | 97.0% | 1963 |
| Serbia | ECA | 2019 | 84.2% | 1059 |
| North Macedonia | ECA | 2018/19 | 90.0% | 855 |
| Suriname | LAC | 2018 | 82.1% | 2389 |
| Iraq | MENA | 2018 | 99.1% | 8452 |
| Georgia | ECA | 2018 | 84.7% | 1405 |
| Kosovo | ECA | 2019/20 | 78.2% | 827 |
| Tonga | EAP | 2019 | 96.0% | 734 |
| Lower-Middle-Income | | | | |
| Algeria | MENA | 2018 | 94.5% | 5785 |
| Mongolia | ECA | 2018 | 95.5% | 3485 |
| Tunisia | MENA | 2018 | 96.3% | 1466 |
| Kiribati | EAP | 2018/19 | 98.2% | 1137 |
| Ghana | SSA | 2017/18 | 99.2% | 4788 |
| Sao Tome and Principe | SSA | 2019 | 97.9% | 1065 |
| Zimbabwe | SSA | 2018/19 | 96.1% | 3473 |
| Bangladesh | SA | 2019 | 93.0% | 11,671 |
| Lesotho | SSA | 2018 | 87.5% | 1845 |
| Kyrgyz Republic | ECA | 2018 | 98.4% | 1901 |
| Nepal | SA | 2019 | 98.4% | 3901 |

**Table 1.** *Cont.*

| Country | Region | Year of Survey | Under 5 Response Rate | Sample Size |
|---|---|---|---|---|
| Low-Income | | | | |
| Guinea-Bissau | SSA | 2018/19 | 99.1% | 3987 |
| The Gambia | SSA | 2018 | 96.1% | 5302 |
| Chad | SSA | 2019 | 99.4% | 11,006 |
| Togo | SSA | 2017 | 96.4% | 2660 |
| Madagascar | SSA | 2018 | 94.1% | 6617 |
| DR Congo | SSA | 2017/18 | 99.8% | 10,305 |
| Sierra Leone | SSA | 2017 | 99.6% | 6119 |
| Central African Republic | SSA | 2018/19 | 96.7% | 4366 |
| Total | | | | 114,475 |

EAP—East Asia and Pacific; ECA—Europe and Central Asia; LAC—Latin America and Caribbean; MENA—Middle East and North Africa; SA—South Asia; SSA—Sub-Saharan Africa. Sample sizes are unweighted and for children aged 2–4 with valid disability data.

### 2.1. Child Disability and Significant Cognitive Delay

MICS6 contained a new module for 2–4-year-old children to identify children with disabilities. Child disability had been measured in MICS2–4 by an optional module including the Ten Questions Screen (TQS) [21]. However, this measure was dropped by UNICEF following MICS4 due to concerns about: (a) the over-identification of disability associated with the functional domains included in the TQS; (b) the omission of items related to key functional domains such as mental health and psychosocial functioning; (c) the TQS's inability to determine severity of disability; (d) the inapplicability of the TQS to older children; and (e) the lack of cognitive testing of TQS items [22].

The new module, developed by the Washington Group on Disability Statistics (WGDS: http://www.washingtongroup-disability.com/) (accessed on 16 July 2021), was based on informant report (primarily maternal report) of their child's difficulties compared with children of the same age in nine different functional domains (seeing, hearing, walking, fine motor, understanding, being understood, learning, playing, and controlling behavior). Four response options were available for all domains other than the behavior domain; (1) 'no difficulty', (2) 'some difficulty', (3) 'a lot of difficulty', (4) 'cannot do at all'. The controlling behavior domain had five response options; (1) 'not at all', (2) 'less', (3) 'the same', (4) 'more' or (5) 'a lot more'.

Initial validation of the new module (undertaken in three low/middle-income countries) estimated that using the cut-off recommended by the WGDS (primarily based on the child having at least 'a lot of difficulty' in at least one domain) resulted in a prevalence of child disability that ranged from 1.1% in Serbia to 2.0% in Mexico among children aged 2–4 years [23]. We used the cut-off recommended by the WGDS to define child disabilities and child disabilities associated with the specific functional limitations listed above. For all disability measures, the reference group was children without disabilities. Disability data were missing for <1% of children.

We also identified 3–4-year-old children with evidence of significant cognitive delay. Following the procedures used in previous research [12,24], we defined significant cognitive delay as the failure to attain any of the five developmental milestones contained in the literacy-numeracy and learning domains of the Early Child Development Index (ECDI) [25]. All items are based on key informant (primarily maternal) report with simple binary (yes/no) response options.

- Literacy-numeracy: Can the child: (1) identify/name at least ten letters of the alphabet; (2) read at least four simple, popular words; (3) name and recognize the symbols of all numbers from 1 to 10?
- Learning: Can the child: (4) follow simple directions on how to do something correctly; (5) when given something to do, do it independently?

### 2.2. Other Child Demographics

Data were collected on the child's age in years and sex. No data were missing.

*2.3. Indicators of Household Circumstances*

2.3.1. Household Wealth

Household wealth is likely to be associated with the prevalence of child disability [15,23,26]. MICS6 data includes a within-country wealth index for each household. To construct the wealth index, principal components analysis is performed by using information on the ownership of consumer goods, dwelling characteristics, water and sanitation, and other characteristics that are related to the household's wealth, to generate weights for each item. Each household is assigned a wealth score based on the assets owned by that household weighted by factors scores. The wealth index is assumed to capture underlying long-term wealth through information on the household assets [27,28]. These data were collected in all countries. From these data, we created a binary variable; whether the child was living in the poorest 20% of households in their country vs. not. Data were missing for <1% of children.

2.3.2. Urban/Rural Location

Data were released with a within-country defined binary indicator of urban/rural location for each household. No data were missing.

2.3.3. Household Composition

Information was collected on whether the child's natural mother and father were living in the child's home. These data were missing for <1% of children.

2.3.4. Maternal/Caretaker Characteristics

Level of maternal education is also likely to be associated with the prevalence of child disability [23,26]. The highest level of education received by the child's mother was recorded using country-specific categories. We recoded these data into a binary measure; none or only primary education vs. secondary or higher-level education. These data were collected in all countries. Data were missing for <1% of children.

Caretaker disability was determined for caretakers aged 18–49 using the Washington Group Short Set of Questions on Disability (WGSSQD: http://www.washingtongroup-disability.com/) (accessed on 16 July 2021). The module is based on informant report of difficulties in six different functional domains (seeing, hearing, walking, remembering/concentrating, self-care, and communicating). Four response options were available for each domain ('no difficulty', 'yes—some difficulty', 'yes—a lot of difficulty', and 'cannot do at all'). Disability is defined by the WGDS as having 'a lot of difficulty' in one or more domains. Disability data were missing for 6.8% of caretakers aged 18–49.

Data were collected on maternal age in years at the time of interview. We recoded these data into a binary variable; mother under 18 at birth of the target child vs. not. These data were missing for 4.5% of mothers.

2.3.5. Parenting Practices

Information was collected on a range of parenting practices including the child's support for learning and the use of violent parental discipline. Respondents were asked 'In the past 3 days, did you or any household member over 15 years of age engage in any of the following activities with (name): (a) read books to or looked at picture books with (name)?; (b) told stories to (name)?; (c) sang songs to (name) or with (name), including lullabies?; (d) took (name) outside the home, compound, yard or enclosure?; (e) played with (name)?; (f) named, counted, or drew things to or with (name)?' We defined support for learning as an adult having engaged in four or more activities to promote learning and school readiness in the past 3 days. Respondents were also asked 'How many children's books or picture books do you have for (name)?' and 'I am interested in learning about the things that (name) plays with when he/she is at home. Does he/she play with: (a) homemade toys (such as dolls, cars, or other toys made at home)? (b) toys from a shop or manufactured toys? (c) household objects (such as bowls or pots) or objects found outside (such as sticks,

rocks, animal shells or leaves)?'. An adequate number of books was defined as having three or more children's books (MICS indicator TC.50). An adequate number of playthings was defined as having two or more playthings (MICS indicator TC.51). These two items were combined into a single item of having adequate books and having adequate playthings. We defined low child stimulation as the presence of either low support for learning or inadequate books and playthings in the home. Data were missing for <1 of children.

A Child Discipline module was included in MICS, adapted from the Parent–Child Conflict Tactics Scale [29]. Respondents were told, 'All adults use certain methods to teach children the right behavior or address a behavior problem. I will read various methods that are used, and I want you to tell me whether you or anyone else in your household has used each method with (child's name) in the last month.' The respondents then answered No (0) or Yes (1) to whether they or any other adults in their household had used each of eight forms of aggressive or violent discipline: 'shouted, yelled or screamed at child'; 'called child dumb, lazy or another name'; 'shook child'; 'spanked, hit or slapped child on bottom with bare hand'; 'hit or slapped child on the hand, arm or leg'; 'hit child on the bottom or elsewhere with belt, brush, stick, etc.'; 'hit or slapped child on the face, head or ears'; 'beat child up as hard as one could'. From these data, we constructed a binary variable of violent parental discipline based on the reported use of any of the three most violent forms of parental discipline; hitting the child with a weapon, hitting the child on the face/head or beating up the child 'as hard as one could' [14]. Data were missing for <1% of children.

### 2.3.6. Attitudes toward Intimate Partner Violence

Data were collected from mothers on their attitudes toward social acceptability of intimate partner violence (IPV) by the following questions; 'Sometimes a husband/partner is annoyed or angered by things that his wife does. In your opinion, is a husband/partner justified in hitting or beating his wife in the following situations: [A] If she goes out without telling him? [B] If she neglects the children? [C] If she argues with him? [D] If she refuses to have sex with him? [E] If she burns the food?' These were recoded into a binary variable social acceptability of IPV (yes to any of the five scenarios vs. no to all scenarios. Data were missing for the caretakers of 5.0% of children.

### 2.4. Child Access to Pre-School Education

Information was collected on whether the enumerated child was currently attending any form of pre-school education. In most countries, these questions were only asked of children age 3–4 years. These data were missing for 14.5% of 3–4-year-old children, with particularly high rates of missing data in Sierra Leone (89.6%), Guinea Bissau (84.1%), Chad (43.6%) and Lesotho (43.7%).

### 2.5. Country Characteristics

Given the commonly reported association between child well-being and national wealth in low- and middle-income countries [30], we used World Bank 2018 country economic classification groups of upper-middle-income, lower-middle-income and low-income [31]. These classifications are based on per capita Gross National Income adjusted for purchasing power parity (pcGNI; expressed as current US$ rates) using the World Bank's Atlas Method. No data were missing.

### 2.6. Approach to Analysis

In the first stage of analysis, we used simple bivariate descriptive statistics to estimate the prevalence of exposure of children with/without disabilities and with/without significant cognitive delay to each indicator of home circumstances (with 95% confidence intervals) for each country economic classification group and an overall pooled estimate. In the second stage of analysis, we used multilevel modelling to investigate the extent to which risk of exposure to each indicator of home circumstances varied within countries by

child disability after any between group differences in child age and sex were taken into account (adjusted prevalence rate ratio).

All analyses were undertaken using Stata 16 (StataCorp, College Station, TX, USA). Prevalence estimates used the svyset/svy commands to take account of the use of within country cluster sampling. Multilevel mixed effects modelling of within-country associations was undertaken using the mepoisson command to generate prevalence rate ratios adjusted for child age and sex, with the association between child disability status and outcomes being allowed to vary between countries (i.e., treated as a random effect). Given the small amount of missing data on most indicators, complete case analyses were undertaken.

## 3. Results

### 3.1. Prevalence of Child Disability

The prevalence of child disability was 3.05% (95%CI 2.69–3.46%) in upper-middle-income countries, 4.06% (3.67–4.49%) in lower-middle-income countries and 7.92% (7.48–8.40%) in low-income countries. As signified by the lack of overlap of confidence intervals, all differences between country economic classification groups were statistically significant.

### 3.2. Association between Child Disability, Significant Cognitive Delay and Home Circumstances

The overall association between child disability, significant cognitive delay and indicators of home circumstances is presented in Tables 2–4 separately for upper-middle-income countries, lower-middle-income countries and low-income countries.

**Table 2.** The association between child disability and significant cognitive delay with indicators of home circumstances in upper-middle-income countries.

| Indicator | Overall Prevalence for Children with Disability | Overall Prevalence for Children without Disability | Adjusted Prevalence Rate Ratio | Overall Prevalence for Children with Developmental Delay | Overall Prevalence for Children without Developmental Delay | Adjusted Prevalence Rate Ratio |
|---|---|---|---|---|---|---|
| Living in poorest 20% of households | 25.7% (23.9–27.5) (*n* = 1644) | 22.4% (21.6–23.2) (*n* = 16,626) | 1.36 *** (1.20–1.54) (*n* = 18,270) | 39.1% (32.6–45.9) (*n* = 630) | 22.3% (20.6–24.2) (*n* = 16,487) | 1.55 *** (1.36–1.76) (*n* = 17,117) |
| Living in rural area | 42.2% (34.6–50.2) (*n* = 1644) | 38.2% (35.8–40.8) (*n* = 16,626) | 1.06 (0.92–1.22) (*n* = 18,270) | 40.7% (33.7–48.1) (*n* = 630) | 39.0% (36.1–41.9) (*n* = 16,487) | 1.29 *** (1.15–1.45) (*n* = 17,117) |
| Natural mother living in household | 97.5% (95.8–98.6) (*n* = 1644) | 97.2% (96.2–98.6) (*n* = 16,621) | 1.00 (0.93–1.08) (*n* = 18,265) | 99.3% (97.5–99.8) (*n* = 630) | 97.2% (96.3–97.9) (*n* = 16,487) | 1.02 (0.94–1.11) (*n* = 17,108) |
| Natural father living in household | 73.6% (65.1–80.7) (*n* = 1637) | 81.4% (76.0–85.8) (*n* = 16,580) | 0.97 (0.89–1.05) (*n* = 18,217) | 89.6% (79.2–95.1) (*n* = 629) | 81.4% (76.5–85.4) (*n* = 16,429) | 1.00 (0.92–1.09) (*n* = 17,058) |
| Mother has no or only primary education | 19.0% (14.3–24.8) (*n* = 1639) | 14.8% (11.6–18.7) (*n* = 16,596) | 1.17 (0.95–1.45) (*n* = 18,235) | 28.4% (19.8–38.9) (*n* = 629) | 14.9% (11.7–18.8) (*n* = 16,456) | 1.90 ** (1.30–2.76) (*n* = 17,085) |
| Mother/caretaker has a disability | 13.1% (9.2–18.3) (*n* = 1613) | 3.8% (3.3–4.4) (*n* = 16,180 | 2.80 *** (2.15–3.64) (*n* = 17,793) | 3.5% (2.1–5.7) (*n* = 618) | 4.2% (3.7–4.9) (*n* = 15,301) | 0.88 (0.21–3.74) (*n* = 15,919) |
| Mother under 18 at birth of target child | 2.8% (1.3–5.8) (*n* = 1614) | 3.3% (2.9–3.7) (*n* = 16,211) | 0.64 (0.39–1.03) (*n* = 17,825) | 4.6% (2.6–8.2) (*n* = 619) | 3.4% (2.9–3.9) (*n* = 15,981) | 0.73 (0.47–1.15) (*n* = 16,600) |
| Low stimulation | 45.2% (38.6–50.9) (*n* = 1638) | 40.3% (38.6–42.0) (*n* = 16,592) | 1.23 * (1.00–1.52) (*n* = 18,230) | 74.9% (70.2–79.1) (*n* = 626) | 39.9% (38.0–41.8) (*n* = 16,455) | 1.46 * (1.08–1.98) (*n* = 17,081) |
| Violent parental discipline | 26.0% (21.9–30.5) (*n* = 1643) | 19.5% (18.3–20.8) (*n* = 16,625) | 1.29 *** (1.13–1.47) (*n* = 18,268) | 16.7% (6.5–36.7) (*n* = 630) | 11.4% (8.4–15.3) (*n* = 16,486) | 0.76 (0.23–2.46) (*n* = 17,116) |
| Social acceptability of IPV | 25.6% (19.3–33.0) (*n* = 1502) | 26.0% (22.0–30.4) (*n* = 14,532) | 1.17 * (1.02–1.36) (*n* = 16,034) | 46.3% (38.2–54.6) (*n* = 600) | 26.2% (22.4–30.4) (*n* = 14,567) | 1.07 (0.95–1.22) (*n* = 15,167) |
| Child currently attending pre-school education | 33.6% (24.8–43.7) (*n* = 1162) | 50.6% (46.3–54.9) (*n* = 11,622) | 0.87 (0.72–1.05) (*n* = 12,824) | 23.8% (16.5–33.0) (*n* = 618) | 50.4% (46.0–54.7) (*n* = 15,952) | 0.64 * (0.45–0.91) (*n* = 16,570) |

* *p* < 0.05, ** *p* < 0.01, *** *p* < 0.001. All sample sizes (*n*) are unweighted.

**Table 3.** The association between child disability and significant cognitive delay with indicators of home circumstances in lower-middle-income countries.

| Indicator | Overall Prevalence for Children with Disability | Overall Prevalence for Children without Disability | Adjusted Prevalence Rate Ratio | Overall Prevalence for Children with Developmental Delay | Overall Prevalence for Children without Developmental Delay | Adjusted Prevalence Rate Ratio |
|---|---|---|---|---|---|---|
| Living in poorest 20% of households | 25.7% (23.9–27.5) (*n* = 1888) | 22.4% (21.6–23.2) (*n* = 34,125) | 1.16 ** (1.06–1.28) (*n* = 36,013) | 33.2% (30.5–36.0) (*n* = 1755) | 21.2% (20.4–22.1) (*n* = 25,345) | 1.50 *** (1.31–1.71) (*n* = 27,100) |
| Living in rural area | 60.8% (57.1–64.4) (*n* = 1888) | 56.8% (55.5–58.2) (*n* = 34,125) | 1.05 (0.98–1.13) (*n* = 36,013) | 67.1% (63.9–70.1) (*n* = 1755) | 55.7% (54.3–57.1) (*n* = 25,345) | 1.18 ** (1.07–1.30) (*n* = 27,100) |
| Natural mother living in household | 91.0% (89.2–92.6) (*n* = 1887) | 93.9% (93.5–94.3) (*n* = 34,116) | 1.01 (0.96–1.06) (*n* = 36,003) | 93.4% (91.8–94.6) (*n* = 1753) | 93.1% (92.6–93.5) (*n* = 25,324) | 1.00 (0.95–1.05) (*n* = 27,077) |
| Natural father living in household | 68.2% (64.9–71.3) (*n* = 1874) | 77.5% (75.8–79.2) (*n* = 33,935) | 0.97 (0.91–1.03) (*n* = 35,809) | 75.2% (71.9–78.2) (*n* = 1737) | 76.5% (74.6–78.3) (*n* = 25,150) | 1.01 (0.96–1.07) (*n* = 26,887) |
| Mother has no or only primary education | 42.4% (39.3–45.6) (*n* = 1887) | 32.0% (29.2–35.0) (*n* = 34,123) | 1.15 * (1.03–1.28) (*n* = 36,010) | 50.5% (45.3–55.7) (*n* = 1755) | 32.1% (29.0–35.4) (*n* = 25,342) | 1.41 *** (1.32–1.51) (*n* = 27,097) |
| Mother/caretaker has a disability | 10.9% (8.6–13.8) (*n* = 1775) | 3.7% (3.4–4.1) (*n* = 32,971) | 2.12 *** (1.64–2.75) (*n* = 34,746) | 5.0% (3.8–6.4) (*n* = 1637) | 3.9% (3.6–4.3) (*n* = 23,641) | 1.30 * (1.04–1.62) (*n* = 25,287) |
| Mother under 18 at birth of target child | 3.6% (2.7–4.9) (*n* = 1765) | 3.7% (3.1–4.3) (*n* = 32,901) | 1.10 (0.86–1.42) (*n* = 34,666) | 3.9% (2.9–5.3) (*n* = 1628) | 3.7% (3.1–4.5) (*n* = 23,536) | 0.88 (0.68–1.14) (*n* = 25,191) |
| Low stimulation | 66.6% (63.0–70.0) (*n* = 1866) | 52.7% (48.4–56.9) (*n* = 33,778) | 1.09 ** (1.03–1.16) (*n* = 35,644) | 64.4% (58.7–69.7) (*n* = 1734) | 51.1% (46.5–55.7) (*n* = 25,148) | 1.21 *** (1.14–1.28) (*n* = 26,882) |
| Violent parental discipline | 42.8% (39.7–46.1) (*n* = 1887) | 31.9% (28.7–35.3) (*n* = 34,116) | 1.14 ** (1.05–1.23) (*n* = 36,003) | 36.1% (31.9–40.5) (*n* = 1755) | 35.1% (31.4–39.0) (*n* = 25,341) | 0.96 (0.88–1.05) (*n* = 27,096) |
| Social acceptability of IPV | 35.5% (32.2–39.1) (*n* = 1539) | 30.0% (28.7–31.3) (*n* = 29,984) | 1.06 (0.97–1.17) (*n* = 31,523) | 40.5% (37.3–43.8) (*n* = 1412) | 29.4% (28.0–30.8) (*n* = 21,052) | 1.25 *** (1.12–1.40) (*n* = 22,464) |
| Child currently attending pre-school education | 43.4% (39.0–48.0) (*n* = 1188) | 42.4% (41.3–43.5) (*n* = 21,568) | 0.73 ** (0.60–0.89) (*n* = 22,756) | 22.3% (19.6–25.2) (*n* = 1669) | 43.6% (42.6–44.7) (*n* = 24,108) | 0.53 *** (0.44–0.64) (*n* = 25,777) |

* *p* < 0.05, ** *p* < 0.01, *** *p* < 0.001. All sample sizes (*n*) are unweighted.

**Table 4.** The association between child disability and significant cognitive delay with indicators of home circumstances in low-income countries.

| Indicator | Overall Prevalence for Children with Disability | Overall Prevalence for Children without Disability | Adjusted Prevalence Rate Ratio | Overall Prevalence for Children with Developmental Delay | Overall Prevalence for Children without Developmental Delay | Adjusted Prevalence Rate Ratio |
|---|---|---|---|---|---|---|
| Living in poorest 20% of households | 25.7% (23.9–27.5) (*n* = 4289) | 22.4% (21.6–23.2) (*n* = 36,290) | 1.04 (0.93–1.17) (*n* = 40,579) | 30.4% (28.3–32.7) (*n* = 7632) | 21.8% (20.4–23.3) (*n* = 29,460) | 1.27 *** (1.22–1.33) (*n* = 37,092) |
| Living in rural area | 72.0% (68.1–75.6) (*n* = 4289) | 67.2% (63.7–70.6) (*n* = 36,290) | 1.04 (0.98–1.10) (*n* = 40,579) | 77.3% (74.2–80.1) (*n* = 7632) | 66.0% (62.5–69.4) (*n* = 29,460) | 1.11 *** (1.06–1.17) (*n* = 37,092) |
| Natural mother living in household | 89.8% (88.5–91.0) (*n* = 4289) | 89.4% (88.7–90.1) (*n* = 36,285) | 1.00 (0.97–1.04) (*n* = 40,574) | 89.6% (88.4–90.6) (*n* = 7629) | 88.7% (87.9–89.4) (*n* = 29,447) | 1.00 (0.98–1.03) (*n* = 37,076) |
| Natural father living in household | 68.0% (66.0–69.9) (*n* = 4278) | 69.8% (68.8–70.7) (*n* = 36,221) | 0.97 (0.93–1.01) (*n* = 40,499) | 70.7% (68.9–72.4) (*n* = 7618) | 70.0% (69.0–70.9) (*n* = 29,399) | 1.00 (0.97–1.04) (*n* = 37,017) |
| Mother has no or only primary education | 79.2% (76.8–81.3) (*n* = 4289) | 75.4% (73.7–77.0) (*n* = 36,282) | 1.02 (0.98–1.05) (*n* = 40,571) | 81.4% (79.2–83.4) (*n* = 7630) | 75.3% (73.5–77.0) (*n* = 29,453) | 1.08 *** (1.04–1.11) (*n* = 37,083) |
| Mother/caretaker has a disability | 12.0% (10.5–13.7) (*n* = 4081) | 4.5% (4.1–4.9) (*n* = 34,988) | 2.30 *** (1.78–2.97) (*n* = 39,069) | 6.3% (5.5–7.3) (*n* = 6881) | 4.9% (4.5–5.4) (*n* = 26,645) | 1.12 (1.00–1.25) (*n* = 33,526) |
| Mother under 18 at birth of target child | 8.1% (7.1–9.3) (*n* = 4068) | 6.6% (6.3–6.9) (*n* = 34,942) | 1.14 * (1.00–1.29) (*n* = 39,010) | 7.0% (6.2–7.8) (*n* = 6856) | 6.5% (6.2–6.9) (*n* = 26,509) | 1.04 (0.94–1.15) (*n* = 33,365) |

**Table 4.** *Cont.*

| Indicator | Overall Prevalence for Children with Disability | Overall Prevalence for Children without Disability | Adjusted Prevalence Rate Ratio | Overall Prevalence for Children with Developmental Delay | Overall Prevalence for Children without Developmental Delay | Adjusted Prevalence Rate Ratio |
|---|---|---|---|---|---|---|
| Low stimulation | 68.4% (65.8–70.8) (*n* = 4242) | 71.2% (70.0–72.4) (*n* = 35,888) | 0.97 (0.91–1.04) (*n* = 40,130) | 77.7% (75.8–79.5) (*n* = 7539) | 72.0% (70.7–73.3) (*n* = 29,131) | 1.10 *** (1.07–1.13) (*n* = 36,670) |
| Violent parental discipline | 40.0% (37.8–42.3) (*n* = 4289) | 39.0% (37.9–40.1) (*n* = 36,284) | 1.08 (0.99–1.19) (*n* = 40,573) | 39.1% (37.0–41.2) (*n* = 7613) | 42.9% (41.8–44.0) (*n* = 29,457) | 0.91 ** (0.86–0.97) (*n* = 37,088) |
| Social acceptability of IPV | 67.0% (64.6–69.4) (*n* = 3995) | 59.1% (57.6–60.5) (*n* = 34,256) | 1.11 ** (1.04–1.17) (*n* = 38,251) | 64.7% (62.2–67.1) (*n* = 6717) | 59.5% (58.1–61.0) (*n* = 26,052) | 1.06 (0.99–1.13) (*n* = 32,742) |
| Child currently attending pre-school education | 8.6% (6.9–10.6) (*n* = 2381) | 13.7% (12.2–15.4) (*n* = 18,327) | 0.86 (0.74–1.01) (*n* = 20,708) | 3.8% (3.0–4.7) (*n* = 5774) | 15.7% (14.0–17.5) (*n* = 21,446) | 0.37 ** (0.21–0.67) (*n* = 27,220) |

\* $p < 0.05$, ** $p < 0.01$, *** $p < 0.001$. All sample sizes (*n*) are unweighted.

Children with disabilities and children with significant cognitive delay were more likely than their peers to be living in relative household poverty in all three country economic classification groups. The level of increased risk was statistically significant for all but one comparison (children with disabilities in low-income countries). In all three country economic classification groups, the level of increased risk of living in relative poverty was notably higher for children with significant cognitive delay than for children with disabilities.

Children with disabilities and children with significant cognitive delay were also more likely than their peers to be living in rural areas in all three country economic classification groups. However, the level of increased risk was only statistically significant for children with significant cognitive delay.

There were no statistically significant associations between children with disabilities or children with significant cognitive delay and any indicator of household composition (mother/father living in the child's household). There was only one statistically significant association between maternal age (under 18) at birth and child disability/significant cognitive delay; in low-income countries, mothers under 18 at the birth of the target child were more likely to have a child with disabilities.

Children with disabilities and children with significant cognitive delay were more likely than their peers to have a mother with only primary level education in all three country economic classification groups. The level of increased risk was statistically significant for all but two comparisons (children with disabilities in upper-middle- and low-income countries). It is worth noting, however, that the effect size for increased risk among children with disabilities in upper-middle-income countries was similar to the statistically significant effect size for increased risk among children with disabilities in lower-middle-income countries. In all three country economic classification groups, the level of increased risk of having a mother with only primary level education was notably higher for children with significant cognitive delay than for children with disabilities.

Children with disabilities and children with significant cognitive delay were more likely than their peers to have a mother with disabilities in five of the six comparisons across the three country economic classification groups (the exception being children with significant cognitive delay in upper-middle-income countries). The level of increased risk was statistically significant for all but two of the six comparisons (children with significant cognitive delay in upper-middle- and low-income countries). In all three country economic classification groups, the level of increased risk of having a mother with disabilities was notably higher for children with disabilities than for children with significant cognitive delay.

Children with disabilities and children with significant cognitive delay were more likely than their peers to receive low levels of parental stimulation in five of the six comparisons across the three country economic classification groups (the exception being children with disabilities in low-income countries). The level of increased risk was statistically sig-

nificant for all but one comparison (children with disabilities in low-income countries). In all three country economic classification groups, the level of increased risk of being exposed to low levels of stimulation was notably higher for children with significant cognitive delay than for children with disabilities.

Children with disabilities were more likely than their peers to be exposed to violent parental discipline in all three country economic classification groups. The level of increased risk was statistically significant for two of the three comparisons (the exception being children with disabilities in low-income countries). In contrast, children with significant cognitive delay were less likely than their non-disabled peers to be exposed to violent parental discipline in all three country economic classification groups, although these differences were only statistically significant in one instance (children with significant cognitive delay in low-income countries).

Children with disabilities and children with significant cognitive delay were more likely than their peers to live in households where intimate partner violence was considered acceptable by the child's mother in all three country economic classification groups. The level of increased risk was statistically significant for three of the six comparisons.

Children with disabilities and children with significant cognitive delay were less likely than their peers to be receiving any pre-school education in all three country economic classification groups. The level of increased risk was statistically significant for all but two of the six comparisons (children with disabilities in upper-middle- and low-income countries). In all three country economic classification groups, the level of increased risk of not receiving pre-school education was notably higher for children with significant cognitive delay than for children with disabilities.

## 4. Discussion

Our analyses of the home circumstances of over 100,000 children with/without disabilities or significant cognitive delay across 30 middle- and low-income countries indicated that:

- Across all three country economic classification groups, children with disabilities and children with significant cognitive delay were more likely than their peers to be living in relative household poverty and in rural areas, to have a mother with only primary level education, to live in households where intimate partner violence was considered acceptable by the child's mother and to be less likely to be receiving any pre-school education;
- In the majority of country economic classification groups, children with disabilities and children with significant cognitive delay were more likely than their peers to have a mother with disabilities and to receive low levels of parental stimulation;
- Across all three country economic classification groups, children with disabilities were more likely than their peers to be exposed to violent parental discipline;
- For many of these indicators of poorer home circumstances (e.g., household poverty, rural location, low maternal education, and low stimulation), the level of risk was notably higher for children with significant cognitive delay than for children with disabilities.

Our study adds to knowledge about the home circumstances of children with disabilities and children with significant cognitive delay in four important ways. First, with data collected between 2017 and 2020, it provides relatively contemporary information on the living circumstances of children under 5 years of age with either disabilities or significant cognitive delay in a wide range of middle- and low-income countries.

Second, sophisticated cluster sampling strategies with extremely high survey response rates and low levels of missing data (except in the case of access to pre-school education) combine to give high confidence that the resulting data are representative of the national populations of children under 5 years of age in the participating countries.

Third, the study, to our knowledge, provides novel data on the exposure of young children with either disabilities or significant cognitive delay to some indicators of poorer home

circumstances (e.g., living in households where intimate partner violence was considered acceptable by the child's mother).

Finally, by disaggregating analyses by child disabilities more generally and significant cognitive delay, the results provide a more nuanced picture of the risk of exposure to some adverse childhood experiences among the broad population of children with disabilities.

### 4.1. Implications

Our study findings suggest the need to consider tailored, evidence-based approaches to reduce and (potentially) overcome the increased risks that young children with disabilities and young children with significant cognitive delay experience in all three country classifications (upper- and lower-middle-income and low-income countries). To ensure these young children can fulfill their developmental potential, interventions are required to both (i) prevent/avert damaging home circumstances likely to arise in families with children with disabilities/significant cognitive delay (e.g., violent parental discipline, intimate partner violence), and (ii) reduce/overcome current deleterious home circumstances where the healthy growth, development and well-being of children with disabilities/significant cognitive delay is already compromised.

The increased risks for children with disabilities and children with significant cognitive delay are likely to arise from multiple levels of influence—individual, relationship, organization, policy and society. Emerson et al. [12], for example, have shown that a combination of organizational, policy and individual relationship (home) interventions in early childhood in low- and middle-income countries (every mother having secondary-level education; every household having access to improve water and sanitation; and every child having an acceptable level of home stimulation) would reduce the loss of developmental potential among children in low- and middle-income countries. Interventions tailored to cultural understanding of childhood disability and disability more broadly and particularly when these are focused on the family home have a much greater chance of success [32].

Policy interventions such as identifying children with disabilities/significant cognitive delay in the first 1000 days and subsequently prioritizing pre-school education for these children show promising results, as do policies with a particular focus on households in rural areas and/or low maternal education [3,6,7]. In addition, our results highlight the need for interventions to be targeted at families supporting children with disabilities/significant cognitive delay living in rural areas and the need for ensuring that interventions are fit for purpose when supporting families facing challenging circumstances.

### 4.2. Limitations

The results of our study need to be considered in light of a number of limitations. As with all large general national health and social surveys, the time available for data collection on any particular issue is limited. As such, short, abbreviated scales or modules are often employed to measure key constructs. For example, in the present study limitations are evident in the measures employed to identify maternal disability and significant cognitive delay. Regarding the former, the use of the WGSSQD to identify disability has been criticized for underrepresenting adults with disabilities associated with mental health problems [33]. Regarding the latter, concerns have been expressed about the sensitivity of the ECDI items for detecting developmental delay [34,35]. It should be noted, however, that a revised and extended 20-item version of the ECDI (ECDI2030) has been developed which contains 10 items in the 'learning' domain. Its incorporation in future rounds of MICS surveys is likely to provide a much more robust basis for identifying young children with developmental delay. In addition, questions have been raised regarding potential biases in the identification of child disability that may arise in the new WGDS module on child functioning as a result of the requirement for informants to judge their child's functioning in comparison with children of the same age, a requirement that in the context of national surveys demands a high level of knowledge regarding normative levels of child development in the informants country [24]. Finally, basing analyses on the SCD

status of the first enumerated child will have led to a degree of misclassification of whether households were supporting a child with SCD.

*4.3. Future Research*

The findings of this study point to several lines for further research. The increased risk of having a mother with disabilities for young children with disabilities/significant cognitive delay requires further investigation to unravel the relationship between maternal disability type (functional limitations) and functional limitations of the child. The higher risk for children with disabilities/significant cognitive delay to live in households where intimate partner violence was considered acceptable by the child's mother is worrying and deserves further research attention for two reasons: the solid evidence on the deleterious and sustained impact of intimate partner violence on children [36] and the emerging evidence on impaired cognitive functioning for women subject to ongoing intimate partner violence [37].

**5. Conclusions**

- Adverse family circumstances have a negative impact on the well-being of children. They may also reduce the effectiveness of home-based early intervention programs unless they are specifically tailored to the level of resources available.
- We undertook secondary analysis of nationally representative data collected in 30 middle- and low-income countries involving over 100,000 children with/without disabilities or significant cognitive delay.
- Children with disabilities and/or significant cognitive delay were more likely than their peers to:
  - Be living in relative household poverty and in rural areas;
  - Have a mother with only primary level education;
  - Live in households where intimate partner violence was considered acceptable by the child's mother;
  - Be less likely to be receiving any pre-school education;
  - Have a mother with disabilities;
  - Receive low levels of parental stimulation;
  - Be exposed to violent parental discipline.
- Interventions are needed to reduce the risk of exposure of children with disabilities to these adversities.
- In addition, these aspects of family context need to be considered when designing home-based early intervention programs to support children with disabilities to reach their developmental potential.

**Author Contributions:** Conceptualization, E.E. and G.L.; methodology, E.E. and G.L.; formal analysis, E.E.; writing—original draft preparation, E.E.; writing—review and editing, E.E. and G.L. Both authors have read and agreed to the published version of the manuscript.

**Funding:** This research received no external funding.

**Institutional Review Board Statement:** Ethical approval for each survey was provided by the relevant research ethics approval process in each participating country (see published country reports for full details at https://mics.unicef.org/surveys, accessed on 16 July 2021). Given the data provided by UNICEF had been stripped of all identifying information ethical approval specific to the analyses undertaken in the present paper was deemed unnecessary.

**Informed Consent Statement:** Informed consent was obtained from all subjects involved in the study. Informed consents were handled in each country in which data were collected (see published country reports for full details at https://mics.unicef.org/surveys, accessed on 16 July 2021).

**Data Availability Statement:** Following approval by UNICEF, MICS6 data were downloaded from http://mics.unicef.org/ (accessed on 16 July 2021).

**Acknowledgments:** We would like to thank the UNICEF's global Multiple Indicator Cluster Surveys (MICS) program for allowing us to use the datasets.

**Conflicts of Interest:** The authors declare no conflict of interest.

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
