# Peer review of "The Circumstances of Children with and without Disabilities or Significant Cognitive Delay Living in Ordinary Households in 30 Middle- and Low-Income Countries"

_disabilities, doi:10.3390/disabilities1030014_

Round 1

Reviewer 1 Report

This paper used UNICEF’s Multiple Indicator Cluster Survey to understand household circumstances of roughly 100,000 children with/without disabilities or significant cognitive delays.

This is a well-executed study and contributes to the scholarship.

Major and Minor concerns in order of appearance:

-A bit more connection between home context and how that is important (even two sentences) would be nice in the introduction. For example, how does household circumstances alter the success/execution of home-based intervention programs?

-I understand the need to focus the research on the first enumerated child under 5 in the survey, but it is important to discuss the implications of this, at least in the limitations section.

-It would be helpful to add if the estimates of child disability available in this survey (for each country) align with estimates of child disability provided from another reliable source. This information could possibly be added to table 1 to show if this survey provides a consistent, over-estimate, or under-estimate of disability prevalence. If not in the table, just a bit of information in text.

-How is the missing data addressed in cases with extensive missing (pre-school enrollment)?

-In the measures section there are a variety of cut-off points listed, such as lines 175 -177“we constructed a binary variable…based on the reported use of any of the three most violent forms of discipline. Or lines 160-161 “defined as having three or more children’s books”. Do these (and the other) cut-offs have any theoretical support? Any precedence? If so, citations would support these decisions. If not, there needs to be more explanation about these specific cut-offs.

-It would be helpful to put some of the key coefficients (with significance) in the results section.

-Add sample sizes to the tables

-Typos in the implication section

-Additional limitation: time order- are these prior to child with disability or possibly a result of.

Reviewer 2 Report

The article is a good example of secondary data analysis, and how it can be used for deeper analysis on disability.

I have no specific comments on the text. The premises, approach, methodology and data seem sound to me.

I have to two suggestions:

1. The section "Discussion", especially the section's introductory part and the sub-section "Implications": they are too much packed. I suggest to break them into shorter, independent paragraphs.

2. The way that findings and their implications are outlined in the section "Discussion" are clear to a scholarly readership (though they can made clearer, see suggestion 1). However, I suggest to add a very simple section or sub-section where the authors present a very simple recap of findings, implication, and needs for future research. Something light, for instance in the form of bullet points and written in a simple, lay language. This would help the non-technical readership understand the value of the analysis. It could also be used by the authors and the journal to promote the article on social media and to various stakeholders.
